# Characterization of Alginate Utilization Strategy in a Novel Marine *Bacteroidetes*: Insights from *Roseihalotalea indica* gen. nov. sp. nov. TK19036^T^

**DOI:** 10.3390/md23090334

**Published:** 2025-08-24

**Authors:** Zheng Fu, Shunqin You, Defang Wu, Runying Zeng, Kai Tang, Zhuhua Chan

**Affiliations:** 1Technical Innovation Center for Utilization of Marine Biological Resources, The Third Institute of Oceanography, Ministry of Natural Resources, Xiamen 361005, China; fuzheng@tio.org.cn (Z.F.); youshunqin@external.tio.org.cn (S.Y.); zeng@tio.org.cn (R.Z.); 2School of Life Sciences and Biopharmaceutics, Shenyang Pharmaceutical University, Shenyang 117004, China; 3State Key Laboratory of Marine Environmental Science, College of Ocean and Earth Science, Fujian Key Laboratory of Marine Carbon Sequestration, Xiamen University, Xiamen 361005, China

**Keywords:** *Roseihalotalea indica*, alginate lyase, alginate, polysaccharide utilization loci, alginate degradation mechanism

## Abstract

Alginate, a major polysaccharide in brown algae, is vital for the carbon cycling of the ocean ecosystem and holds promise for biotechnological applications. Marine *Bacteroidetes*, known for the ability to degrade complex polysaccharides, play an important role in the ocean carbon cycle; however, the detailed alginate degradation pattern remains to be further explored. In this study, an alginate utilization locus was identified in the genome of a new marine *Bacteroidetes*, *Roseihalotalea indica* gen. nov. sp. nov. TK19036^T^, and encodes two new alginate lyases, RiAlyPL6 and RiAlyPL17, which play potential roles in the degradation and utilization of alginate. RiAlyPL6 and RiAlyPL17 have distinct degradation products and substrate preferences, revealing the adaptation of the strain to utilize alginate with different M/G ratios. Based on the results in this paper, we have proposed a model for the degradation and utilization mechanism of alginate in *Roseihalotalea indica* gen. nov. sp. nov. TK19036^T^. All in all, our research provides a new insight into the alginate mechanisms within marine *Roseihalotalea*, and the two novel alginate lyases are excellent candidates for preparation and application.

## 1. Introduction

The phylum *Bacteroidetes* is among the most abundant phyla in the ocean after *Proteobacteria* and *Cyanobacteria*, contributing about 10–30% of marine surface bacterioplankton [1]. They are distributed globally in a wide range of marine ecosystems. *Bacteroidetes* are recognized for their powerful ability to degrade and utilize complex polysaccharides via various types of polysaccharide utilization loci (PULs), which encode multiple carbohydrate-active enzymes (CAZymes) [2]. Therefore, *Bacteroidetes* play important roles in the ocean organic carbon cycle.

As a crucial nutrient for various marine heterotrophic bacteria, alginate, an anionic linear hetero-polysaccharide, consists of α-L-guluronate (G) and its C-5 epimer, β-D-mannuronate (M), linked through β-1,4-glycoside bonds [3]. These two monosaccharides are arranged as homopolymers, poly-α-L-guluronate (poly-G) and poly-β-D-mannuronate (poly-M), or a random heteropolymer (poly-MG) (Figure 1). As a natural biocompatible polysaccharide from brown algae, alginate is widely used in the food, agriculture, textile, and medical fields [4], and degradation by alginate lyases is one of the important processes in the ocean carbon cycle [5].

Alginate lyase (EC 4.2.2.-), an important marine biological enzyme, can cleave 1,4-glycoside bonds via β-elimination reactions to produce alginate oligosaccharides with unsaturated uronic acid (Δ) at the non-reducing end or 4,5-unsaturated uronic acid monomers (Figure 1) [6]. Based on differences in amino acid sequence similarity, alginate lyases can be classified into 15 polysaccharide lyase (PL) families, including PL-5, -6, -7, -8, -14, -15, -17, -18, -31, -32, -34, -36, -39, -41, and -44 in the Carbohydrate-Active enZYmes (CAZy) database [7,8]. These enzymes can also be divided into poly-G-specific lyase (EC 4.2.2.11), poly-M-specific lyase (EC 4.2.2.3), and bifunctional lyase (EC 4.2.2-), which can act on both [9].

Alginate is a linear polysaccharide consisting of α-L-guluronate (G) and its C-5 epimer β-D-mannurinate (M) connected by 1,4 O-linked glycosidic bonds, which are randomly arranged into different blocks. Alginate lyase degrades alginate via a β-elimination reaction, producing unsaturated C=C double bonds between C4 and C5 at the non-reducing end. AOS: Alginate oligosaccharides; UAOS: Unsaturated alginate oligosaccharides. Figure drawn with ChemBioDraw Ultra 12.0.

Hitherto, three different alginate utilization strategies have been reported in bacteria: the PUL system, the “scattered” system (mainly in *Vibrio* sp.), and the “pit” transport system (only reported in *Sphingomonas* sp. strain A1) [10]. In the PUL system, extracellular alginate lyases degrade high degree polymerization (DP) substrates into alginate oligosaccharides (AOS) and are transported into the periplasm by SusC/SusD-like, where oligoalginate lyases (Oals) catalyze AOS into monomers. Subsequently, monomers are transferred into the cytoplasm via the major facilitator superfamily (MFS) transporter or inner membrane transporter-like protein (ToaABC). The monomer are then catalyzed into 4-deoxy-L-erythro-5-hexoseulose uronate (DEH) by KdgF [11] or spontaneously degraded stepwise into 2-keto-3-dexy-gluconate (KDG) and keto-deoxy-phospho-gluconate (KDPG), which are processed into pyruvate and 3-phosphoglyceraldehyde (G-3-P) via the Entner–Doudoroff (ED) pathway, ultimately giving energy to bacteria. Unlike the PUL system, the “scattered” system comprises related genes scattered throughout the genome instead of an operon and transfers products into the cell through the outer membrane porin KdgMN and the inner membrane symporter ToaABC, but not the SusC/SusD-like and the MFS system [5]. The “pit” system directly incorporates alginate into the cytoplasm through a super-channel consisting of a “mouth-like pit” on the outer membrane [12].

A new strain *Roseihalotalea indica* gen. nov. sp. nov. TK19036^T^ of a novel genus *Roseihalotalea* within the family *Catalinimonadaceae* was isolated and identified from the Southwest Indian Ocean in a previous study [13,14]. This strain is enriched in a diverse repertoire of CAZymes and PULs, which are capable of degrading a variety of polysaccharides. This study investigated Alginate-PUL and characterized two new alginate lyases, RiAlyPL6 and RiAlyPL17, revealing their unique enzymatic properties and their sequential roles and strategies in alginate degradation. Based on the above results, a model of the alginate metabolic pathway in *Roseihalotalea indica* gen. nov. sp. nov. TK19036^T^ was proposed. The findings not only extend the insight into alginate utilization in marine *Roseihalotalea* but also provide valuable resources for alginate lyases preparation and application.

## 2. Results

### 2.1. Analysis of Alginate Utilization in Roseihalotalea indica gen. nov. sp. nov. TK19036^T^

*Roseihalotalea indica* gen. nov. sp. nov. TK19036^T^ was predicted to encode a rich repertoire of CAZymes and PULs. Among them, it contained a PUL that encodes the uptake or degradation of alginate. As shown in Figure 2, there are several genes annotated in this PUL, e.g., WKN34553, a PL-6 alginate lyase; WKN34549, a PL-17 alginate lyase; and six key enzymes involved in monosaccharide catabolism: WKN34546, WKN34545, WKN34544, WKN34543, WKN34542, and WKN34541. Additionally, one complete SusC/SusD gene pair (WKN40357 and WKN34551) and various transcription factors (WKN34554, WKN34552, WKN34548, WKN34547, and WKN34540) were also encoded in this putative Alginate-PUL.

To evaluate the capacity of *Roseihalotalea indica* gen. nov. sp. nov. TK19036^T^ to utilize alginate, a growth experiment was performed with sodium alginate, AOS, and glucose as the sole carbon source, during which the growth of the strain was monitored (Figure 3). When cultured in the medium with glucose, the cells grew rapidly to reach the logarithmic growth phase, and the stable growth began on day 3. Likewise, the growth curve of AOS group was consistent with the glucose group. However, the OD_600_ in the alginate group was maintained at about 0.04 after inoculation, which suggested that *Roseihalotalea indica* gen. nov. sp. nov. TK19036^T^ was unable to maintain its normal growth owing to its inability to use alginate.

### 2.2. Sequence Analysis of RiAlyPL6 and RiAlyPL17

RiAlyPL6 is composed of 748 amino acids and has a theoretical isoelectric point (pI) of 5.07, with a molecular weight (Mw) of 82.9 kDa. Based on the online sequence analysis (SignalP 6.0/BLASTp/CD-search/dbCAN/PSORT), RiAlyPL6 contains a 22-residue Sec/SPII signal peptide (Met1-Cys22) and two PL-6 catalytic modules (Leu29-Asn398 and Lys474-Ile680), which are localized in the cytoplasm.

Meanwhile, RiAlyPL17 is composed of 752 amino acids and its theoretical Mw and pI are 84.77 kDa and 5.38, respectively. RiAlyPL17 contains a 25-amino acid Sec/SPI signal peptide (Met1-His25) and two PL-17 catalytic modules (Gly182-Leu284 and Ile407-Leu538) with unclear localization.

Phylogenetic analysis indicates that RiAlyPL6 is a member of the PL-6 alginate lyase family and RiAlyPL17 belongs to the PL-17 alginate lyase family (Figure 4A). Amino acid conservation showed that RiAlyPL6 (Arg265 and Lys244, Figure 4B) and RiAlyPL17 (Tyr275 and Tyr465, Figure 4C) contained conserved catalytic residues corresponding to the PL-6 and -17 families [15], respectively.

### 2.3. Recombinant Expression of RiAlyPL6 and RiAlyPL17

RiAlyPL6 and RiAlyPL17, both devoid of signal peptides, were recombinantly expressed using pET28a(+) in *E. coli* BL21(DE3) cells and subsequently purified via a HisTrap HP column. The purification parameters are detailed in Table 1. SDS-PAGE analysis revealed that RiAlyPL6 and RiAlyPL17 had molecular weights of approximately 83 kDa and 85 kDa, respectively, indicating significant enrichment within the anticipated molecular weight range (Figure 5).

### 2.4. Functional Characterization of RiAlyPL6 and RiAlyPL17

RiAlyPL6 exhibited optimal activity at 35 °C (Figure 6A), while the enzyme retained ≥85% relative activity after one hour of incubation at 0 to 30 °C (Figure 6C). According to Figure 6B, the optimal pH for RiAlyPL6 was found to be 7.0 in Tris-HCl buffer. After incubation at pH 10.0 to 10.6 for 12 h, RiAlyPL6 maintained over 80% of its initial activity, demonstrating superior stability under neutral pH conditions (Figure 6D). The effects of metal ions, detergents, or chelators on RiAlyPL6 activity are presented in Figure 6E. Ca^2+^ ions significantly increased RiAlyPL6 activity (more than 20 times). As shown in Figure 6F, the concentration of CaCl_2_ exerts significant effects on RiAlyPL6 activity, with maximal activity observed at 1.0 M CaCl_2_, acting as a critical activator. To ascertain the specificity of substrate, enzymatic activity was assayed using 0.30% (*w*/*v*) alginate, poly-M, and poly-G. The results displayed in Figure 6G revealed that RiAlyPL6 exhibited a preference for poly-G, with relative activity values of 8.33 ± 0.17% (sodium alginate), 38.07 ± 0.06% (poly-M), and 100.00 ± 2.27% (poly-G), respectively.

In addition, the optimal temperature and optimal pH of RiAlyPL17 were 30 °C and 6.6, respectively (Figure 7A,B). The stability of RiAlyPL17 was retained over 80% of its peak activity when incubated at 0–30 °C for 1 h or within a pH range of 7.0–7.6 for 12 h (Figure 7C,D). The activity of RiAlyPL17 was significantly increased by Ca^2+^, Mg^2+^, and Mn^2+^ (Figure 7E). The effect of CaCl_2_ concentrations on activity was investigated. RiAlyPL17 maintained above 70% activity from 0–1.0 M CaCl_2_ and showed the highest activity at 1.0 M (Figure 7F). Regarding substrate specificity, RiAlyPL17 demonstrated high substrate specificity for poly-M and low activity for alginate (7.00 ± 0.45%) and poly-G (30.74 ± 1.58%) (Figure 7G).

To examine the mode of action of RiAlyPL6 and RiAlyPL17, aliquots of the enzymatic reaction mixture were withdrawn at different times and used for thin-layer chromatography (TLC) and size-exclusion chromatography (SEC) analyses (Superdex 30 Increase 10/300 GL). In the TLC result presented in Figure 8A, RiAlyPL6 generated multiple low DP oligosaccharides, and RiAlyPL17 produced only a single oligosaccharide during degradation. Likewise, the same results were obtained using SEC. With increasing reaction time, a total of three obvious absorption peaks appeared in RiAlyPL6 (Figure 8B left), but the only unsaturated oligosaccharide absorption peak at 18.1 mL was observed in RiAlyPL17 (Figure 8B right).

To investigate the final products of RiAlyPL6 and RiAlyPL17, the final products were harvested and identified using ESI-MS after 12 h reaction time (Figure 8C). As shown in Figure 8C left, di-, tri-, and tetra-saccharides are the end products of RiAlyPL6, and the only identified product of RiAlyPL17 is an unsaturated monosaccharide (m/z = 174.96, Figure 8C right). Therefore, RiAlyPL6 degrades alginate to unsaturated oligosaccharides with DP 2–4 as the end products, and unsaturated monosaccharide is the main final product of RiAlyPL17.

## 3. Discussion

*Bacteroidetes* is one of the most abundant heterotrophic marine bacteria and plays a key role in the carbon cycle in ocean ecosystems [2]. *Roseihalotalea indica* gen. nov. sp. nov. TK19036^T^ is a new marine *Bacteroidetes* isolated from the southwest Indian Ocean in a previous work [13]. Its genome is enriched with various CAZymes and PULs targeting pectin, fructan, arabinan, arabinoxylan, and alginate, among others [14]. This research paper focuses on the Alginate-PUL.

The Alginate-PUL of *Roseihalotalea indica* gen. nov. sp. nov. TK19036^T^ encodes two alginate lyases (RiAlyPL6 and RiAlgPL17) and a series of monosaccharide metabolism genes (KdgF, DehR, KdgK, and KdgA), which conform to the PUL system for alginate assimilation [10]. The subcellular localization of RiAlyPL6 was predicted to be cytoplasmic, and that of RiAlgPL17 was unknown; however, we prefer to believe that RiAlyPL6 and RiAlgPL17 are both localized in the periplasm. RiAlyPL6 has similar properties to RiAlgPL17 in terms of optimum temperature, optimum pH, and the effect of metal ions, but differs in substrate preferences and degradation patterns. This result indicates that the degradation process of alginate may be performed first by the endo-lyase RiAlyPL6 and subsequently by the exo-lyase RiAlgPL17 in vivo. Meanwhile, RiAlyPL6 preferentially degrades GG blocks in contrast to RiAlyPL17, which prefers to act on the remaining MM blocks. This enzyme combination has been reported as a strategy employed by bacteria for the utilization of alginate with different M/G ratios [5].

*Roseihalotalea indica* gen. nov. sp. nov. TK19036^T^ could use AOS as its sole carbon source; however, it could not use alginate. This result provides supporting evidence for the subcellular localization prediction of the two alginate lyases and indicates that the strain TK19036^T^ is deficient in the ability to cleave alginate due to the lack of extracellular alginate lyase. Recently, three different ecophysiological types of alginate utilization were categorized by Hehemann et al. [16]: pioneer, scavenger, and harvest. Marine *Bacteroidetes* is generally known as harvest in ocean ecosystems during the alginate degradation [17,18]. Given the above empirical results in this paper, *Roseihalotalea indica* gen. nov. sp. nov. TK19036^T^ appears to be a scavenger that can only utilize oligomers, and it is also confirmed that different strategies for alginate degradation are observed in microorganisms belonging to different phyla [10].

Based on the findings of the above research, the metabolism of alginate in *Roseihalotalea indica* gen. nov. sp. nov. TK19036^T^ has been proposed, as illustrated in Figure 9. Briefly, the process occurs as follows:(i)With the help of the SusC/SusD-like transporter, which is located on the outer membrane and delivered into the periplasm, the produced AOS from other microorganisms is converted into monomers via Oals, RiAlyPL6, and RiAlyPL17, regardless of the M/G ratio. RiAlyPL6 cleaves the GG blocks first via an endo-mode, and then RiAlyPL17 cuts on the MM blocks via an exo-mode, while the MG blocks may be degraded relatively slowly.(ii)Monomers are transported into the cytoplasm through the MFS transporter, form DEH by KdgF, and are converted into the final product KDPG via multiple downstream enzymes (DehR, KdgK, and KdgA), which are eventually assimilated by the central metabolic cycle.

However, it remains unknown how the high DP of AOS and poly-MG is utilized by *Roseihalotalea indica* gen. nov. sp. nov. TK19036^T^ and will need further investigations.

## 4. Materials and Methods

### 4.1. Materials and Strains

Sodium alginate (from brown algae) was purchased from Sigma-Aldrich (St. Louis, MO, USA), while poly-M and -G (6–8 kDa) were acquired from Qingdao BZ Oligo Biotech Co., Ltd. (Qingdao, China). AOS were isolated and prepared using VsAly7A as previously described [19]. *Roseihalotalea indica* gen. nov. sp. nov. TK19036^T^ was a kind gift from Kai Tang (Xiamen University, Xiamen, China) and was cultured in 2216E medium (Qingdao Haibo Biotech Co., Ltd., Qingdao, China). *E. coli* BL21 (DE3) (TaKaRa, Dalian, China) was served as the expression host for recombinant proteins expressed in Luria-Bertani (LB) medium with 30 mg/mL kanamycin.

### 4.2. Prediction of Alginate-PUL in Roseihalotalea indica gen. nov. sp. nov. TK19036^T^

The assembled genome sequences weredeposited in the NCBI database (GenBank: CP120682) in previous studies. Initial functional annotation was conducted via the RAST (Rapid Annotation using Subsystem Technology) server (https://rast.nmpdr.org/, accessed on 28 January 2024) [20]. Pfam domains (http://pfam.xfam.org/, accessed on 15 February 2024) [21] and dbCAN3 (http://bcb.unl.edu/dbCAN2/, accessed on 27 February 2024) [22] were combined to identify potential CAZymes genes, and their functions were accurately predicted. Homologous protein sequences were identified via BLASTp searches against the NCBI database. Alginate utilization gene clusters were identified as potential Alginate-PUL. Alginate-utilizing genes, including CAZymes, sugar transporters, and transcriptional regulators, were located adjacent to alginate lyases, which extended the boundary of Alginate-PUL. To find Alginate-PUL, the genome sequences were also used to search and predict substrates using dbCAN CGCFinder [22].

### 4.3. Sole-Carbon-Source Cultivation of Roseihalotalea indica gen. nov. sp. nov. TK19036^T^

Sodium alginate and AOS were chosen as test substrates. Cells seeded were harvested at the logarithmic growth stage via centrifugation (1000 rpm, 30 min), and washed 3 times with sterile artificial seawater (KCl 0.3 g, MgSO_4_·H_2_O 0.5 g, CaCl_2_·H_4_O_2_ 0.038 g, NH_4_Cl 0.3 g, K_2_HPO_4_ 0.3 g, NaCl 35 g, Milli-Q water 1 L, pH 7.8–8.0) before inoculation of seed. Cultivation experiments in a sole-carbon-source medium, designed using 2216E medium as background, were conducted in triplicate at 25 °C for 12 days, with glucose as the control group. Briefly, the organic carbon and nitrogen components of the medium were removed and an appropriate amount of NH_4_Cl was added. The growth of *Roseihalotalea indica* gen. nov. sp. nov. TK19036^T^ was monitored using a spectrophotometer through optical density measurements at 600 nm (OD_600_).

### 4.4. Sequence Analysis of Alginate-Lyase-Encoding Gene

SignalP 6.0 Server (http://www.cbs.dtu.dk/services/SignalP-6.0/, accessed on 8 March 2024) [23] was used to predict the presence of an N-terminal signal peptide in RiAlyPL6 and RiAlyPL17. PSORTb v3.0 (https://www.psort.org/psortb/, accessed on 8 March 2024) [24] was used for subcellular localization prediction. ExPASy ProtParam (https://web.expasy.org/compute_pi/, accessed on 8 March 2024) was utilized to predict the theoretical isoelectric point (pI) and molecular weight (Mw) of RiAlyPL6 and RiAlyPL17. Conserved domain architecture analysis was conducted via NCBI Conserved Domain Database and Conserved Domain Architecture Retrieval Tool (CDART) [25]. Multiple sequence alignments of RiAlyPL6 and RiAlyPL17 orthologs were generated using Clustal Omega version 1.2.2 [26] and ESPript 3.0 [27]. Phylogenetic analysis was performed using the molecular Evolutionary Genetics Analysis (MEGA) program [28], version 11 (https://www.megasoftware.net/, accessed on 8 March 2024), using the maximum likelihood method.

### 4.5. Cloning, Expression, and Purification of RiAlyPL6 and RiAlyPL17

*Roseihalotalea indica* gen. nov. sp. nov. TK19036^T^ genome sequence was used as the basis for designing gene-specific primers with *Nde I/Xho I*, as listed in Table 2. Following the amplification of target gene fragments, they were inserted into the plasmid pET-28a(+) to generate the pET-based expression vectors for RiAlyPL6 and RiAlyPL17.

Chemically competent *E.coli* BL21 (DE3) cells were transformed with recombinant plasmids and propagated in LB medium containing 30 μg·mL^−1^ kanamycin sulfate at 37 °C. Protein expression was induced at an optical density of 0.5 at 600 nm (OD_600_) with 0.02 mmol·L^−1^ isopropyl β-D-1-thiogalactopyranoside (IPTG), followed by incubation at 18 °C for 24 h under shaking conditions at 200 rpm. The proteins were purified via the HisTrap^TM^ HP column (Cytiva, Uppsala, Sweden). Subsequently, 12.5% SDS-PAGE was used to assess the purity and Mw of the proteins, and their concentrations were measured using a bicinchoninic acid (BCA) assay (New Cell & Molecular Biotech Co., Ltd., Suzhou, China).

### 4.6. Enzymatic Activity Assay of RiAlyPL6 and RiAlyPL17

Enzymatic activity assay was performed by incubating serially diluted RiAlyPL6 or RiAlyPL17 (100 μL) with 0.3% (*w*/*v*) standard sodium alginate substrate (900 μL) in 0.02 M Tris-HCl buffer containing 0.1 M CaCl_2_ (pH 8.0). The reaction mixtures were incubated at 30 °C for 10 min, followed by the quantification of 4,5-unsaturated products using a UH5300 spectrophotometer (Hitachi High-Technologies, Tokyo, Japan) at 235 nm. A single unit of lyase activity (U) was defined as the quantity of enzyme that generated a rise of 0.1 A235 units per minute [29].

### 4.7. Biochemical Characterization of RiAlyPL6 and RiAlyPL17

The influence of pH on enzymatic activities was investigated using a range of buffers, specifically 0.02 M Na_2_HPO_4_–citric acid (pH 3.0–8.0), 0.02 M Tris-HCl (pH 7.05–8.95), 0.02 M Na_2_HPO_4_–NaH_2_PO_4_ (pH 6.0–8.0), and 0.02 M glycine-NaOH (pH 8.6–10.6). Additionally, pH stability was assessed in these solvents following a 12 h incubation period at 4 °C. Furthermore, the degradation system was analyzed within a temperature range of 0–60 °C to ascertain the optimal catalytic temperature. The thermostability of these enzymes was assessed by preincubation with a phosphate buffer (0.02 M, pH 7.3) at various temperatures (0–60 °C) for one hour, followed by conducting an enzyme assay at 40 °C. The effect of CaCl_2_ was investigated by examining the enzyme activities in phosphate buffer (pH 8.0) containing various concentrations of CaCl_2_ (0–1.0 M) at the optimal temperature and pH. Relative activity was calculated by comparing it to the maximal enzymatic activity (considered 100%) in these experiments. All assays were performed in triplicate, and enzymatic activity was expressed as mean ± standard deviation (SD).

### 4.8. Analysis of Reaction Pattern and End Products of RiAlyPL6 and RiAlyPL17

Thin-layer chromatography (TLC) and AKTA FPLC chromatography systems (Cytiva) were employed to assess the mode of action. Briefly, a mixture of 10 mL 0.3% (*w*/*v*) standard sodium alginate and 1 mL of RiAlyPL6 or RiAlyPL17 (0.2 U) was incubated under optimized conditions of 1, 5, 10, 20, 30, or 1, 30, 60, 90, 120 min. Degradation products were analyzed using silica gel plates (TLC Silica gel 60 F254, Merck, Beijing, China) and a mobile phase composed of 1-butanol:formic acid:water (4:6:1, *v*/*v*/*v*) [19]. Plates were stained with diphenylamine-aniline-phosphate (DPA) reagent and charred at 130 °C to visualize carbohydrate bands. Size-exclusion chromatography (SEC) analysis was performed using a Superdex 30 Increase 10/300 GL column (Cytiva, Marlborough, MA, USA). Elution profiles were monitored at 235 nm using an AKTA Purifier system (Cytiva) [30]. For mass spectrometric characterization, reaction mixtures were desalted and analyzed using negative-ion electrospray ionization mass spectrometry (ESI-MS) [31].

## 5. Conclusions

In this study, the alginate utilization mechanism of a novel marine *Bacteroidetes, Roseihalotalea indica* gen. nov. sp. nov. TK19036^T^ was clarified. *Roseihalotalea indica* gen. nov. sp. nov. TK19036^T^ carries a gene cluster involved in the degradation, transport, and metabolism of alginate and encodes two new alginate lyases, RiAlyPL6 and RiAlyPL17, which showed distinct substrate preference and degradation products, assisting the strain to utilize different M/G ratio substrates. These results contributed to a better understanding of the alginate utilization process in *Roseihalotalea* and emphasize the ecological importance of the *Bacteroidetes* phylum in the global carbon cycle.

## Figures and Tables

**Figure 1 marinedrugs-23-00334-f001:**
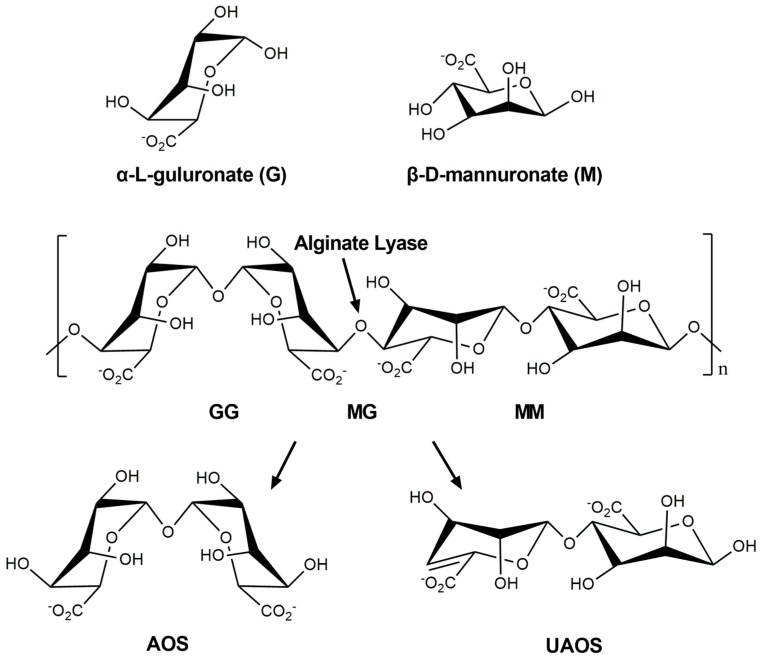
Structures of alginate and degradation by alginate lyase.

**Figure 2 marinedrugs-23-00334-f002:**
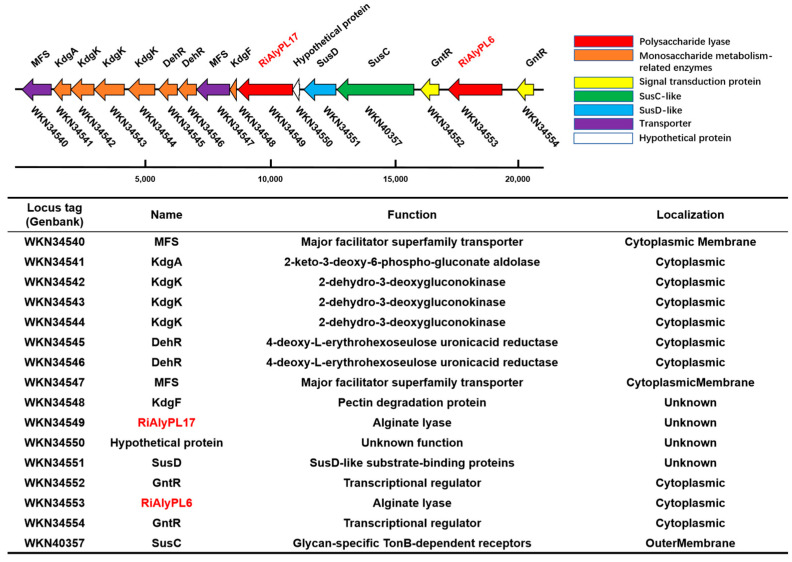
Predicted polysaccharide utilization locus of alginate (Alginate-PUL) in *Roseihalotalea indica* gen. nov. sp. nov. TK19036^T^.

**Figure 3 marinedrugs-23-00334-f003:**
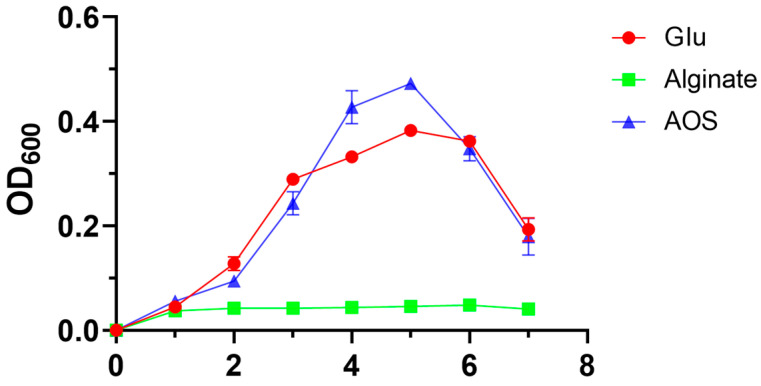
Growth cruves of *Roseihalotalea indica* gen. nov. sp. nov. TK19036^T^ in the medium with glucose, sodium alginate, or AOS as the sole carbon source.

**Figure 4 marinedrugs-23-00334-f004:**
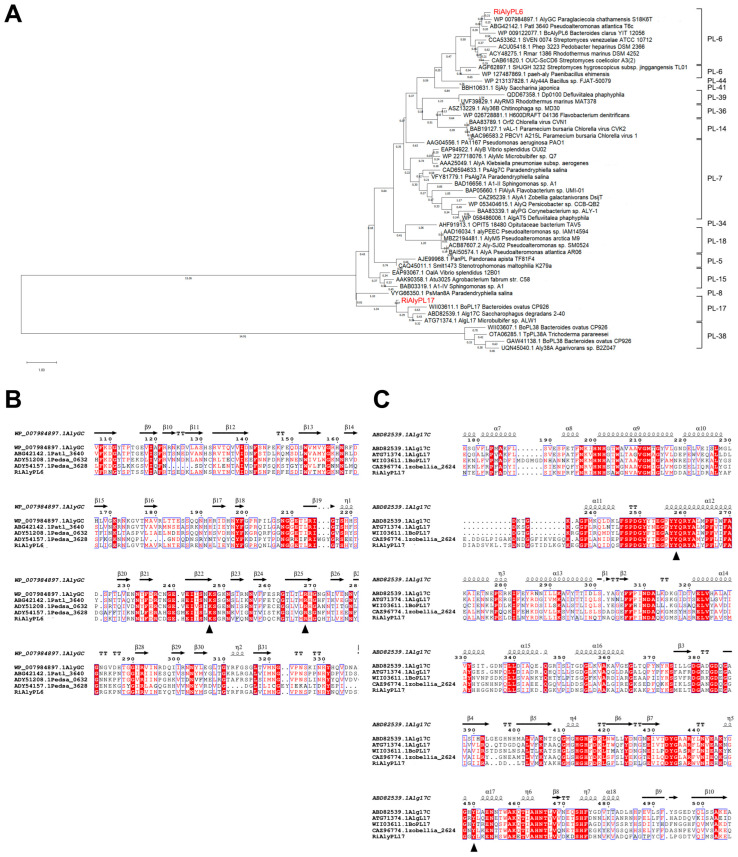
Sequence analysis of the alginate lyases RiAlyPL6 and RiAlyPL17. (**A**) Phylogenetic analysis of RiAlyPL6, RiAlyPL17, and other alginate lyases from different families. (**B**) Comparison of the amino acid sequences of RiAlyPL6 with PL6 alginate lyases. (**C**) Comparison of the amino acid sequences of RiAlyPL6 with PL17 alginate lyases.

**Figure 5 marinedrugs-23-00334-f005:**
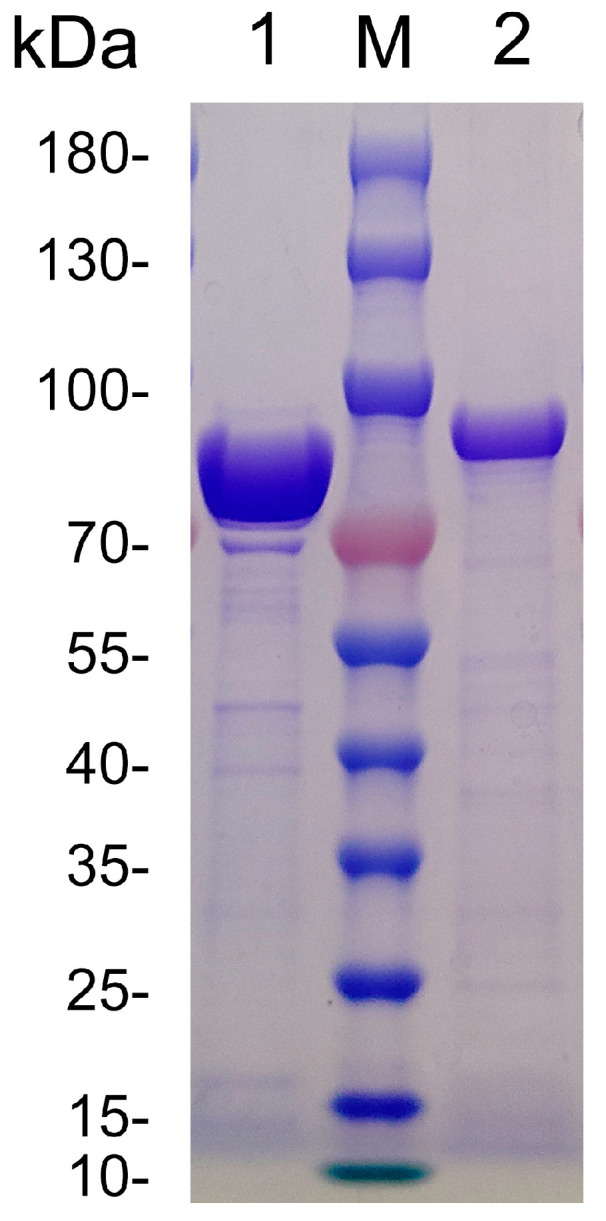
SDS-PAGE analysis of the purified recombinant RiAlyPL6 and RiAlyPL17. Lane M, molecular weight markers. Lane 1, RiAlyPL17. Lane 2, RiAlyPL6.

**Figure 6 marinedrugs-23-00334-f006:**
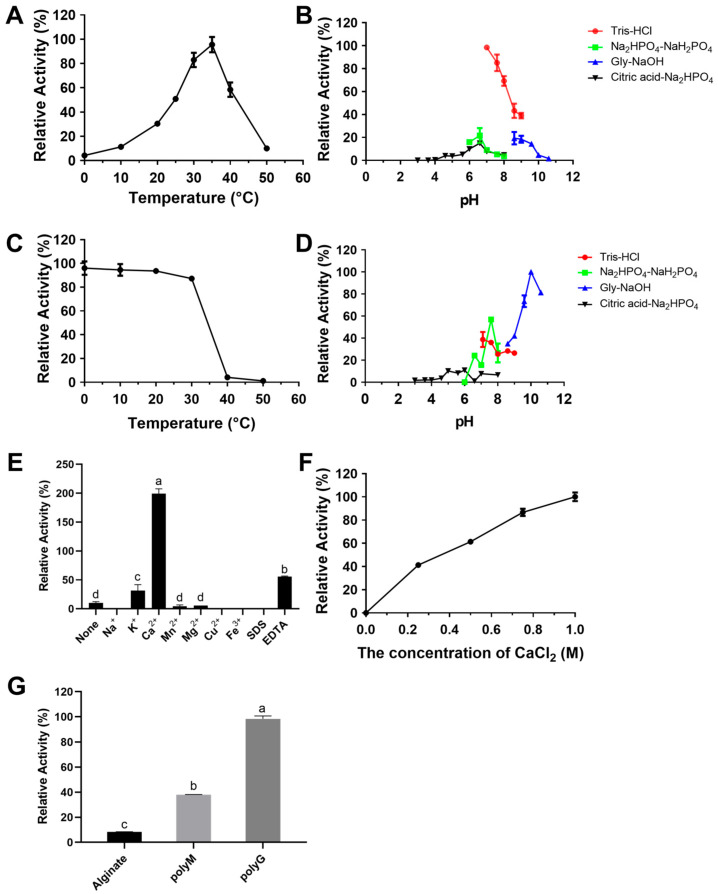
Biochemical characterization of RiAlyPL6. Optimal temperature (**A**), optimal pH (**B**), thermal stability (**C**), pH stability (**D**), effect of metal ions (**E**), effect of CaCl_2_ concentration (**F**), and substrate specificity (**G**) of RiAlyPL6. Values represent the mean of three replicates ± standard deviation, and different letters indicate significant differences at *p* < 0.05 between groups.

**Figure 7 marinedrugs-23-00334-f007:**
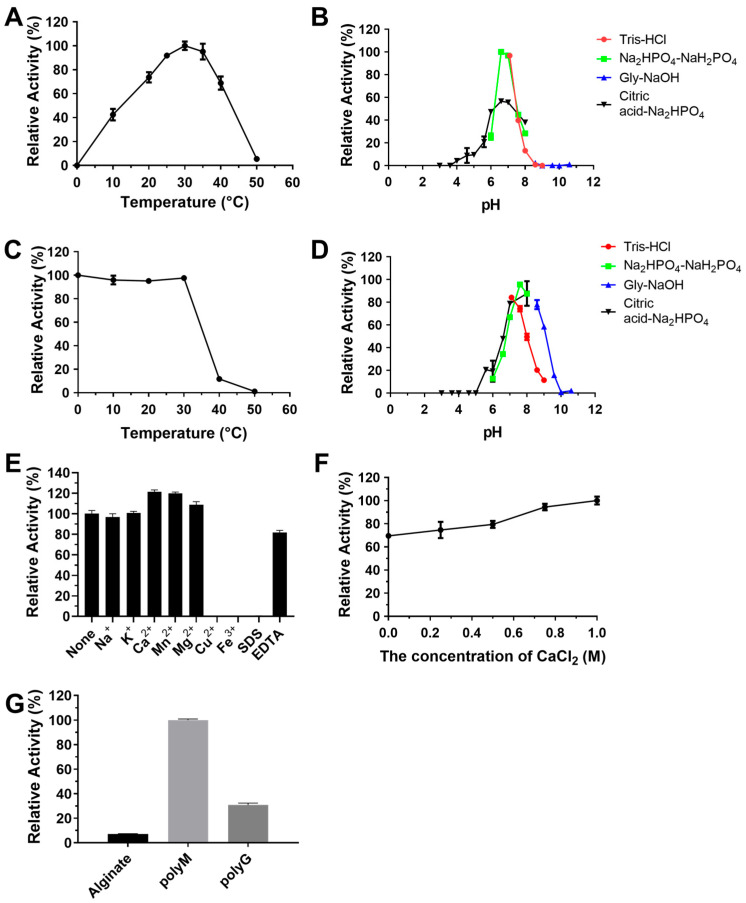
Biochemical characterization of RiAlyPL17. Optimal temperature (**A**), optimal pH (**B**), thermal stability (**C**), pH stability (**D**), effect of metal ions (**E**), effect of CaCl_2_ concentration (**F**), and substrate specificity (**G**) of RiAlyPL17. Values represent the mean of three replicates ± standard deviation.

**Figure 8 marinedrugs-23-00334-f008:**
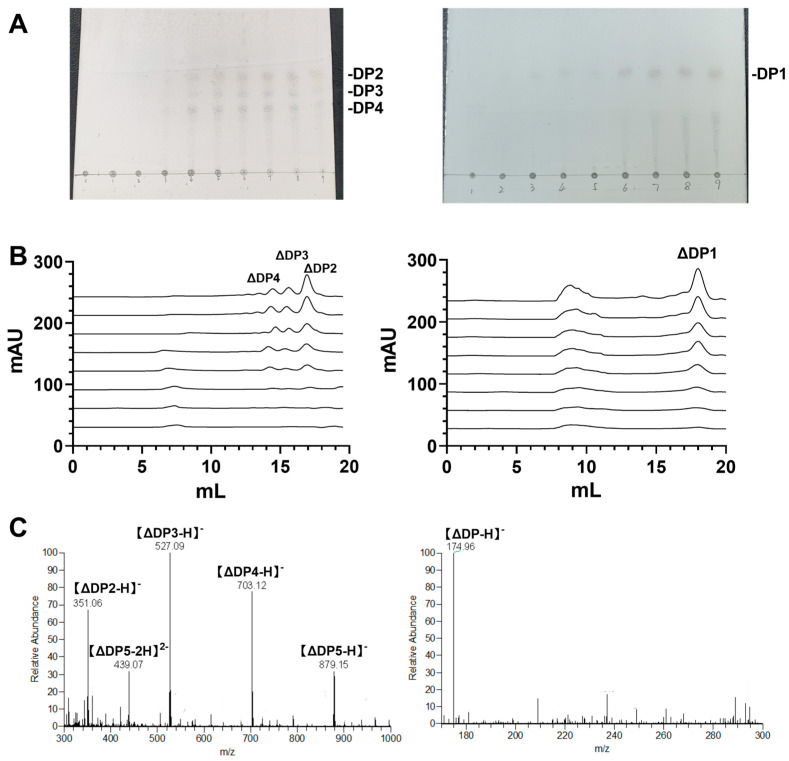
Degradation mode and the end products of RiAlyPL6 and RiAlyPL17. (**A**) The time course of alginate degradation by RiAlyPL6 (**Left**) and RiAlyPL17 (**Right**) using TLC (**A**) and SEC (**B**). (**C**) Analysis of the end products of RiAlyPL6 (**Left**) and RiAlyPL17 (**Right**) using ESI-MS analysis.

**Figure 9 marinedrugs-23-00334-f009:**
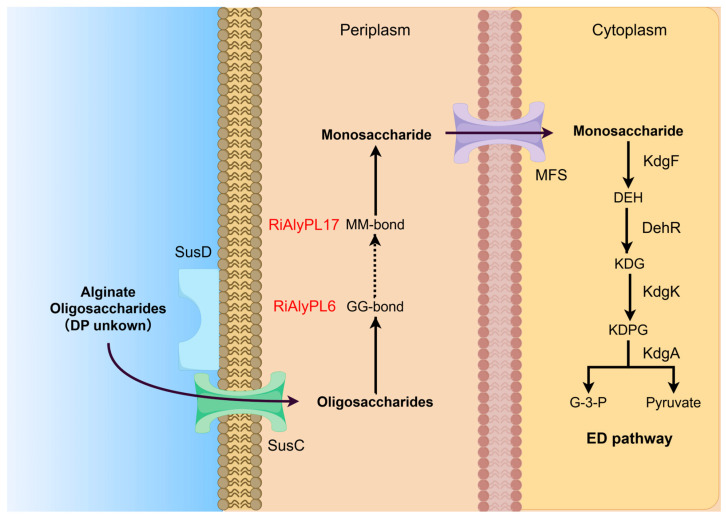
Alginate degradation strategy of *Roseihalotalea indica* gen. nov. sp. nov. TK19036^T^. SusD: SusD-like substrate-binding proteins receptors; SusC: glycan-specific TonB-dependent receptors; MFS: Major facilitator superfamily; KdgF: pectin degradation protein KdgF; DehR: 4-deoxy-L-erythrohexoseulose uronicacid reductase; KdgK: 2-dehydro-3-deoxygluconokinase; KdgA: 2-keto-3-deoxy-6-phospho-gluconate aldolase; DEH: 4-deoxy-L-erythro-5-hexoseulose uronate; KDG: 2-keto-3-dexy-gluconate; KDPG: keto-deoxy-phospho-gluconate; G-3-P: glyceraldehyde triphosphate.

**Table 1 marinedrugs-23-00334-t001:** Purification parameters of recombinant RiAlyPL6 and RiAlyPL17.

Sample		Total Protein (mg)	Total Activity (U)	Specific Activity (U/mg)	Fold	Recovery (%)
RiAlyPL6	Crude enzyme	393.50	22.70	0.58	1.00	100
	Affinity chromatography	53.24	12.56	1.18	2.05	53.30
RiAlyPL17	Crude enzyme	331.95	48.68	0.147	1.00	100
	Affinity chromatography	84.86	20.74	0.233	1.59	42.60

**Table 2 marinedrugs-23-00334-t002:** Primers used in this study.

Primers	Sequence (5′ to 3′)	Usage
RiAlyPL6-F	gtgccgcgcggcagccatatgGATAGTGCCGGCCCCAAT	Expression of RiAlyPL6
RiAlyPL6-R	gtggtggtggtggtgctcgagATCGCTTATAACTTTAACATCTTCTTTATCA	Expression of RiAlyPL6
RiAlyPL17-F	gtgccgcgcggcagccatatgCAGGTGCACCCCAACCTTATT	Expression of RiAlyPL17
RiAlyPL17-R	gtggtggtggtggtgctcgagTTTCTGTCCATTGTCCTTGTGTTT	Expression of RiAlyPL17

## Data Availability

The data presented in this study are available on request from the corresponding author.

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
