# Peer review of "Characterization of Alginate Utilization Strategy in a Novel Marine Bacteroidetes: Insights from Roseihalotalea indica gen. nov. sp. nov. TK19036T"

_marinedrugs, 2025, doi:10.3390/md23090334_

Round 1

Reviewer 1 Report

Comments and Suggestions for Authors

This manuscript, titled "Characterization of Alginate Utilization Strategy in a Novel Marine Bacteroidetes: Insights from Roseihalotalea indica gen. nov. sp. nov. TK19036T" (ID: marinedrugs-3810199), has been carefully reviewed. It stands out as well-written and offers a significant contribution to understanding alginate degradation mechanisms in marine *Bacteroidetes*. The results presented here provide valuable insights that can advance both ecological studies on the marine carbon cycle and biotechnological applications involving alginate.

The identification and biochemical characterization of two novel alginate lyases, RiAlyPL6 and RiAlyPL17, in *Roseihalotalea indica TK19036T*, are particularly compelling. The methods employed to map the genetic machinery and characterize the expressed proteins appear robust and well-executed.

However, there are a few areas where the manuscript could be improved for greater clarity and scientific robustness. Part of the proposed metabolic model, especially what is depicted in Figure 9, seems to venture into more speculative territory. Specific details such as the action of SusC/SusD-like transporters, the exact cleavage sequence of Oals, RiAlyPL6, and RiAlyPL17, and the subsequent transport and processing of monomers via MFS transporters and enzymes like KdgF, DehR, KdgK, and KdgA, would benefit from a more explicit discussion of the supporting evidence within the text.

It is crucial for the authors to more clearly articulate how their experimental findings directly support the complex mechanisms they propose. For instance, while the paper indicates that RiAlyPL6 primarily cleaves GG blocks in an endo-mode and RiAlyPL17 acts on MM blocks in an exo-mode, the justification for this precise sequence and the supposed slower degradation of MG blocks is not fully developed in the current discussion. The excellent biochemical characterization, in fact, already provides strong foundations for these conclusions. RiAlyPL6's preference for polyG substrates and the production of low-DP oligosaccharides point to an endo-mode of action, while RiAlyPL17's high specificity for polyM and the generation of unique oligosaccharides are hallmarks of an exo-mode. The synergistic relationship between these two enzymes, targeting specific alginate motifs, indeed illustrates a clever bacterial strategy to handle the varied M/G ratios of alginate. The slower degradation of MG blocks can be logically inferred from the preferential action of RiAlyPL6 on GG and RiAlyPL17 on MM, and the absence of a specific enzyme for MG would naturally lead to a less efficient breakdown of these segments. Incorporating these details more explicitly into the discussion would significantly strengthen the work's conclusions.

### Furthermore, some specific questions and recommendations deserve attention:

| *   Figure 1, which presents the chemical structure of alginate and its degradation mechanism, would benefit from a more detailed description within the figure legend itself, including also the indication of the software used to generate the chemical structures. |
|---|

*   In lines 58–60, it would be interesting to clarify whether the three bacterial strategies mentioned—the PUL system, the "scattered" system, and the "pit" transport system—are exclusive to alginate degradation or if they are also involved in the breakdown of other polysaccharides.
*   If *Roseihalotalea indica gen. nov. sp. nov. TK19036T* truly possesses a predicted locus for alginate utilization, as presented in this study, the reason why no significant growth was observed on alginate as the sole carbon source (Figure 3; lines 100-107) requires a more in-depth explanation.
*   Considering the visibly significant differences in the biochemical characterization graphs for RiAlyPL6 (Figure 6B, D, E, and G) and RiAlyPL17 (Figure 7b, d, e, and g) regarding parameters such as optimal pH, pH stability, effect of metal ions/chelators, and substrate specificity, it is crucial to include appropriate statistical analyses. This would validate the significance of the observed variations among the groups.
*   Considering the described alginate metabolism in *Roseihalotalea indica gen. nov. sp. nov. TK19036T*, what molecular mechanisms might limit the degradation of alginate oligosaccharides (AOS) with a high degree of polymerization (DP), and how does this impact the efficiency of alginate utilization by this bacterium?
*   The manuscript lacks appropriate references for the methods used. It is essential to include citations and corresponding references for all methodologies described throughout the text.

Despite these points, the manuscript is, overall, a valuable contribution to understanding alginate degradation mechanisms in marine *Bacteroidetes*, and the results presented offer insights that can significantly advance ecological research on the marine carbon cycle and biotechnological applications of alginate.

Author Response

Dear Reviewers:

We appreciate the opportunity to revise our manuscript titled "Characterization of Alginate Utilization Strategy in a Novel Marine Bacteroidetes: Insights from Roseihalotalea indica gen. nov. sp. nov. TK19036T" and are grateful for the insightful comments provided by the Reviewers. A revised version of the manuscript has been generated where the Reviewers' comments have been taken into account. Those comments are all valuable and very helpful for revising and improving our paper. Our replies to the Reviewers' comments are reported hereafter. All changes with respect to the previous submission have been marked in red in the Word file of the manuscript as Supporting Information for Review Only. The revised manuscript has been renamed as "Manuscript Revision".

Reviewer 2 Report

Comments and Suggestions for Authors

I have some comments as below,

Line 32, contributing about 10-30% …?

Line 39, please indicated that α-L-guluronate (G) is the C-5 epimer of β-D-mannuronate (M).

Line 46, can act on …?

Line 49, suggestion: “in” → “and”

Lines 80-82, the sentence was difficult to be understood, please re-organize it for better understanding.

Line 89, encodes functional proteins that uptake and degrade of alginate? Please check the words!

Line 106, generally, an A600 value ranging 0.1-1.0 is reliable.

Figure 8, in my opinion, the data of an HPLC analysis is enough, and the TLC result is not necessary.

Moreover, I am interested in another question besides the endo- or exo-type mode of alginate lyase, are there and 1H-NMR-data of the final oligosaccharide products to indicate their structure properties or data to indicate protein structure properties?   

Author Response

(The authors gave the same response as above.)

Round 2

Reviewer 2 Report

Comments and Suggestions for Authors

Line 17, a suggestion, “however” is more logically than “but”.

Line 17, Abstract, “mechanism” seemed to be a problem to be solved by the paper, and it remained unclear and undiscovered, e.g., lacking innovations. Only degradation pattern can be applied here!

Line 20, “play”, do the authors have experimental data? e.g., transcriptome or RT-PCT of targeting genes? Only “play potential roles” can be used here!

Lines 47 to 49, can cleave … to produce … at the nonreducing end.

Line 57, “Structures of alginate and the degradation by alginate lyase”. For the “mechanism” will be miss-leading.

Line 62, “Figure drawn with ChemBioDraw Ultra 12.0” can be omitted.

Line 84, “investigated”? “characterized”? Please check the tense problems throughout the manuscript.

Line 94, “contained”

Line 103, “Putative”?

Figure 5, the photo is unfocused and can be replaced.

Line 168, “Different letters indicate a significant difference …”? Please check the grammar.

Line 353, “mechanism” seemed to be undiscovered while only the alginate degradation pattern is half-provided for lacking the photo of yielding unsaturated oligosaccharides visually and directly.  

Comments on the Quality of English Language

Please check the tense and  grammar problems throughout the manuscript.

Author Response

(The authors gave the same response as above.)

Round 3

Reviewer 2 Report

Comments and Suggestions for Authors

The manuscript can be language-edited by a language cooperation or an English speaker.